# Dissecting the Role of Autophagy-Related Proteins in Cancer Metabolism and Plasticity

**DOI:** 10.3390/cells12202486

**Published:** 2023-10-19

**Authors:** Liliana Torres-López, Oxana Dobrovinskaya

**Affiliations:** Laboratory of Immunology and Ionic Transport Regulation, Biomedical Research Centre, University of Colima, Av. 25 de Julio #965, Villas de San Sebastián, Colima 28045, Mexico; oxana@ucol.mx

**Keywords:** autophagy, autophagy-related (ATGs) genes/proteins, cancer plasticity, cancer cell metabolism, aerobic glycolysis, Warburg effect, fatty acid oxidation (FAO), tumor microenvironment

## Abstract

Modulation of autophagy as an anticancer strategy has been widely studied and evaluated in several cell models. However, little attention has been paid to the metabolic changes that occur in a cancer cell when autophagy is inhibited or induced. In this review, we describe how the expression and regulation of various autophagy-related (ATGs) genes and proteins are associated with cancer progression and cancer plasticity. We present a comprehensive review of how deregulation of ATGs affects cancer cell metabolism, where inhibition of autophagy is mainly reflected in the enhancement of the Warburg effect. The importance of metabolic changes, which largely depend on the cancer type and form part of a cancer cell’s escape strategy after autophagy modulation, is emphasized. Consequently, pharmacological strategies based on a dual inhibition of metabolic and autophagy pathways emerged and are reviewed critically here.

## 1. Introduction

Autophagy is a constitutive and highly conserved catabolic process, involving degradation of damaged intracellular material during nutrient deprivation or metabolic stress [1]. Three types of autophagy have been described: macro-autophagy, micro-autophagy and chaperone-mediated autophagy (CMA). Although these three pathways differ from each other, they converge in the degradation of cytoplasmic material mediated by lysosomal enzymes [2]. Macro-autophagy (hereafter referred to as autophagy), is the most-studied autophagy type, tightly regulated by a complex machinery involving multiple autophagy-related (ATGs) genes/proteins [1,3,4,5,6,7].

Autophagy is known to be involved in the pathophysiology of many human diseases, including neurodegenerative [8], cardiovascular [9,10] and autoimmune [11,12] diseases and cancer [13]. To understand the mechanisms of its regulation in mammalian cells, various modulators of autophagy, both activators and inhibitors, were developed and used [14]. Some of these modulators have been proposed for use in anticancer chemotherapy [13].

It should be mentioned here that, in cancer, autophagy plays a controversial role, sometimes acting as a suppressor or as a promoter of tumorigenesis [4,7,15,16,17]. Whether autophagy will be protective or, on the contrary, associated with the death of cancer cells largely depends on the type of cancer as well as the tumor microenvironment, disease stage and external stimuli [1,17]. For each specific situation, the level of autophagy optimal for the survival of tumor cells can be determined. Accordingly, new strategies are being developed for cancer therapy through genetic and pharmacological modulation of autophagy, including either its inhibition [4,6,17,18,19] or induction [6]. Some drugs such as chloroquine (CQ) or hydroxychloroquine (HCQ) alone [13] or in combination with other drugs [4] that inhibit autophagy are in different phases of clinical trials. Likewise, other ATG inhibitors have demonstrated effectiveness in pre-clinical trials [13].

It is well established that the regulation of autophagy is tightly coupled to intracellular metabolic pathways. Briefly, high concentrations of nutrients and metabolites act as a negative regulator of autophagy, while the limitation of ATP or essential nutrients such as glucose or amino acids or the increase in metabolites such as fatty acids and ammonia act as autophagy-inducing signals. During stress conditions (e.g., nutrient and energy deprivation, hypoxia, redox stress, mitochondrial damage), autophagy-derived metabolites support multiple biosynthetic pathways and contribute to energy production. In particular, autophagy-mediated protein catabolism yields free amino acids as building blocks for protein synthesis or substrate to be utilized in the tricarboxylic acid (TCA) cycle for energy production, or as substrate for glucose production through gluconeogenesis. Fatty acids, produced by lipophagy, are converted into acetyl-CoA and fuel the TCA cycle. Several comprehensive reviews, providing detailed information on the regulation of cellular-metabolism-dependent autophagy, are highly recommended [3,4,7,20,21].

Accordingly, under stressful conditions, cancer cells rely on autophagy as a mechanism that allows the simultaneous elimination of damaged organelles and recycling of metabolic blocks for survival and proliferation [1,4].

A number of metabolic alterations and adaptations, which occur in cancer cells to meet all needs of rapid proliferation, apparently differentiate them from healthy cells, and the regulation of autophagy is also different [22]. One of the best-studied metabolic features of cancer cells is the Warburg effect, also known as aerobic glycolysis, in which tumor cells use glycolysis for ATP production despite oxygen availability [23]. Non-oxidative breakdown of glucose allows the cell to produce ATP more rapidly. Although a “glycolytic” phenotype is common in many types of aggressive tumors, they can shift their metabolism towards oxidative phosphorylation (OXPHOS) to survive and progress when glucose availability is limited [24,25]. Thus, the type of metabolism (glycolysis vs. OXPHOS) is not fixed during the progression of most tumors and can be changed at different stages of the disease, which determines the plasticity. Besides energy production, both the aerobic glycolysis and TCA pathways ensure building blocks are available to support the biosynthetic requirements for rapid cell growth [23,25]. Moreover, mitochondria control redox and calcium homeostasis and govern cell death mechanisms, which is crucial for chemoresistance [25]. In addition to TCA, mitochondria also host fatty acid oxidation (FAO), which is up-regulated in many tumors and is linked to survival, stemness, metastasis and drug resistance [26,27]. Thereby, re-programmed cellular metabolism, which ensures cancer cell plasticity, i.e., the ability to survive adverse conditions and avoid cell death after drug treatment, is based on finely regulated cross-talk between different metabolic pathways, including aerobic glycolysis, OXPHOS, FAO and autophagy [4]. Thus, the relationship between autophagy and cancer cell metabolism should be considered in therapeutic strategies, based on the modulation of autophagy, to prevent undesired effects associated with the resistance.

In this review, we summarized the effects of autophagy (ATG) modulations on cellular metabolism and their impact on tumors’ drug resistance.

## 2. Mechanism of Autophagy in Mammals

Intra- and extracellular triggers of autophagy are diverse and include nutrient deficiency or deprivation, growth factor depletion, energy starvation, organelle/DNA damage, infection and hypoxia. As a basis for self-recycling and cell survival in adverse environments, autophagy must be well regulated and controlled [21,28] and goes through a series of steps to ensure correct processing. Although autophagy is mostly a cytoprotective mechanism, excessive self-degradation can lead to type 2 cell death (autophagic cell death) [29,30].

The main feature that distinguishes autophagy from CMA and micro-autophagy is the formation of autophagosomes, spherical double-membrane structures in which the cytoplasmic content (cargo) is encapsulated. The means of encapsulation and the cargo content largely determine whether autophagy is selective or non-selective [31,32]. In nonselective autophagy, the cytoplasmic material is randomly sequestrated in response to starvation, while in selective autophagy, the cargo can be mitochondria, endoplasmic reticulum, protein aggregates or lysosomes, giving the specific name to this autophagosome-mediated recycling process [31,32,33].

The general mechanism by which the autophagic process develops is divided into different steps, as shown in Figure 1 (top panel). For each step in the sequence, a complex and highly regulated machinery is required, which is briefly described below.

### 2.1. Induction

Under metabolic stress, two cellular sensors are capable of recognizing the nutrient and energy status, mammalian target of rapamycin complex 1 (mTORC1) and adenosine monophosphate-activated protein kinase (AMPK) [19]. The ULK1 (Unc-51-like kinase 1) complex is formed by the serine/threonine protein kinase ULK1/2, RB1 inducible coiled-coil 1 (RB1CC1, best known as FIP200), ATG13 and ATG101. The Class III PI3K (phosphatidylinositol 3-kinase) complex contains ATG14, Beclin-1 (BECN1), the catalytic (PI3KC3) and regulatory (PI3KR4) subunit of PI3K. PI3KC3 and PI3KR4 are better known as hVPS34 and hVPS15, respectively.

Under normal conditions, the interaction between the ULK1 complex and mTORC1 keeps ULK1 [34,35] and ATG13 [34] phosphorylated and inactive, whereas under metabolic stress, mTORC1 dissociates from the ULK1 complex, leaving it to interact with the class III PI3K complex. Under low energy conditions, AMPK directly phosphorylates and inhibits mTORC1 [36], activates the class III PI3K complex by phosphorylating hVPS34 and BECN1 [37] and phosphorylates ULK1 [35,38,39], thus connecting energy sensing to mitophagy [38]. All these pathways result in autophagy induction.

### 2.2. Phagophore Nucleation

After autophagy induction, the next step is recruitment of ATGs to a subcellular compartment known as the phagophore assembly site (PAS), where nucleation of an isolated membrane from pre-existing organelles [40] is highly dependent on phosphatidylinositol 3-phosphate (PtdIns3P) and ATG9 [41]. ULK1 regulates ATG9 localization [39,42], in addition to phosphorylating BECN1 and activating PI3KC3/hVPS34 [43], which leads to PtdIns3P formation.

### 2.3. Phagophore Elongation

This step occurs with ATG9 in collaboration with microtubule-associated protein 1 light chain 3 (LC3) in its lipidated form (LC3-II) and the ATG12-ATG5-ATG16L complex. The measurement of LC3-II expression is one of the most frequently used assays to monitor autophagy [44]. The LC3 conversion is dependent on ATG3, ATG7, the ATG12-ATG5-ATG16L complex and ATG4; the last one is regulated by ULK1 [45]. The ATG12-ATG5-ATG16L complex formation is orchestrated by ATG7 and ATG10. The correct assembly of this complex as well as the formation of LC3-II is essential for autophagosome biogenesis, elongation and maturation [46,47], and for LC3-II, the following steps of the autophagic process are essential [47].

### 2.4. Cargo Sequestration

Cytosolic material, designated for degradation, must be engulfed in the phagophore membrane to be sealed later, giving rise to the formation of the phagosome, with its cargo ready for lysosome degradation. For this, several autophagy receptors such as p62/SQSTM1 (sequestosome 1), NIX (BNIP3L) or neighbor of BRCA1 gene 1 (NBR1) are necessary to recruit cytosolic cargo in the collaboration with LC3-II acting as a bridge [48,49,50,51]. Participating receptors will depend on the nature of the cargo [49,50], especially for selective autophagy.

### 2.5. Fusion with Lysosome

The newly formed autophagosome can fuse with a lysosome (autophagolysosome) for cargo degradation. For this process, the outer and inner autophagosome membranes must be degraded so that the cargo is delivered to the lysosome, and they can be degraded by lysosomal hydrolases. The formation of the autophagolysosome is highly regulated by several proteins, including the SNARE (soluble NSF attachment protein receptor) complex and mTORC1 [52,53].

## 3. Cancer Progression under the Control of ATGs

### 3.1. ATG Status in Different Types of Cancer Cells May Vary

BECN1 is one of the most important ATGs in the initial stages of autophagy, being a part of the class III PI3K complex. Although BECN1 mutations have been identified in gastric and colorectal cancers [54], their occurrence in the general cancer landscape is extremely rare [55]. However, BECN1 was considered a haploinsufficient tumor suppressor gene because its monoallelic deletion is frequently seen in breast, ovarian and prostate cancers [56,57,58]. Recent review of the published data on the status of BECN1 in various tumors suggests that its reduced expression often occurs in tumors as compared to normal tissues and that lower BECN1 levels generally correlate with a poorer prognosis [59]. In particular, down-regulation of BECN1 has been reported in lymphoma, melanoma, osteosarcoma and brain and lung cancers [60,61,62,63,64].

Lower expression of key *ATGs* (*BECN1*, *ATG3*, *ATG5*, *ATG4*, *ATG14*) was shown in a large panel of primary acute myeloblastic leukemia (AML) patients as compared to normal granulocytes [65]. Similarly, *p62*/*SQSTM1* mRNA levels are down-regulated in the immature myeloid phenotype (AML cell lines, primary CD34^+^ progenitors cells and primary blasts from AML patients) compared with mature granulocytes from healthy donors [66].

Bone marrow (BM) aspirates from AML patients show decreased *NIX* (*BNIP3L*) mRNA expression in comparison to BM aspirates from healthy controls [67]. Also, in BM samples from AML patients, low mRNA levels of *BECN1* and *p62* indicate worse overall survival. In addition, down-regulation of the ATGs *BECN1*, *LC3* and *NBR1* is shown in AML patients compared to patients with hematological diseases (anemia, thrombocytosis) [68]. Similarly, *LC3*, *ATG5* and *ATG10* are down-regulated in whole-blood samples from complete remission patients compared to newly diagnosed AML patients [69]. Therefore, ATG levels can be used as biomarkers to differentiate more aggressive or resistant phenotypes in leukemias.

However, up-regulation of *BECN1* has also been reported in hematologic malignances. Higher levels of *BECN1* and *ATG5* compared to healthy patients were reported in samples derived from patients with chronic lymphoblastic leukemia (CLL) and AML. However, they were associated with favorable prognosis in CLL, in contrast to AML [70,71]. In CD34+ hematopoietic stem cells from chronic myeloblastic leukemia (CML), *BECN1* and *ATG5* are also overexpressed, along with ATG4 [72]. ATG12 was up-regulated in samples of glucocorticoid-resistant pediatric pre-B acute lymphoblastic leukemia (preB-ALL) in comparison to sensitive ones [73]. High *ATG7* levels in leukemic blasts were associated with shorter remission duration in AML patients [74]. *ATG7* silencing enhanced the sensitivity of AML cells to the chemotherapeutic agents cytarabine and idarubicin in in vitro assays as well as in a mouse model of human AML [74]. Therefore, the role of autophagy and ATGs is apparently different in chronic and acute leukemias of different lineages, which differ by proliferation rates and metabolic profiles.

A higher BECN1 level was detected in colorectal and gastric carcinomas as compared to normal mucosal cells, irrespective of invasion, metastasis and stage [75]. On the other hand, ATG12 was identified as possible biomarker since it is up-regulated in oral squamous cell carcinoma tissues without correlation of the mRNA levels with clinical parameters [76]. Finally, ULK1, BECN1, ATG3, ATG5, ATG7, ATG9, ATG10, ATG12, LC3B and p62/SQSTM1 are expressed in gastric cancer tissues, where differences in expression correlate with clinicopathological characteristics such as histological types and lymph node metastasis, positioning these genes as potential biomarkers in gastric cancer [77].

In this sense, the dysregulation of these ATGs represents an attractive strategy to stop the progression of cancer.

### 3.2. ATG Regulation Is Related to Cell Metabolism

BECN1 is regulated by the anti-apoptotic protein Bcl-2 (B-cell lymphoma 2), forming the BECN1-Bcl-2 complex under normal nutrient conditions. This association prevents BECN1 from participating in autophagy induction, while in nutrient deprivation, Bcl-2 phosphorylation dissociates Bcl-2 from BECN1, leaving BECN1 free to bind to the class III PI3K complex [78].

As mentioned above, many cancer cells meet their energy needs through aerobic glycolysis. Glucose starvation has been shown to induce autophagy through a mechanism involving hexokinase 2 (HK2) binding to mTOR and its subsequent inhibition [79]. It is noteworthy that some autophagic steps are regulated by glycolytic enzymes (Figure 2), which can be related to cancer cell pathophysiology.

BECN1 can be regulated by the glycolytic enzymes pyruvate kinase (PKM2), phosphoglycerate kinase 1 (PGK1) and lactate dehydrogenase A (LDHA). PKM2, a key mediator for the Warburg effect [80], increases the phosphorylation of BECN1 and contributes to cell survival via autophagic activation in a nucleophosmin-mutated AML cell line, where *PKM2* knockdown reduces the phosphorylation of BECN1 at T119 [81]. Similarly, PGK1 directly phosphorylates BECN1 at S30, which in human glioblastoma samples correlate with more aggressive tumors and lower median survival [82]. On the other hand, LDHA-BECN1 colocalization suggests the possible involvement of LDHA in the initiation of cytoprotective autophagy [83]. Finally, in a model of doxorubicin-resistant gallbladder cancer cells, *PGK1* knockdown reduces ATG5 and ATG12 protein levels [84].

The involvement of BECN1 in the induction of autophagy seems to also be related to lipid metabolism through fatty acid β-oxidation (FAO) and subsequent entry into the TCA cycle in myeloid leukemia [85] and colon [86] and gastric [87] cancer.

Additionally, different metabolic profiles have been reported for AML cell lines. NB-4 and HL-60 display the glycolytic phenotype, associated with AKT-mTORC1 activation and low autophagic flux, while KG-1 and THP-1 exhibit preferential OXPHOS, constitutive activation of AMPK-mTORC1 and high autophagic flux [88,89].

### 3.3. ATG Status in Cancer Microenvironment

Tumors represent a specific type of pathologic tissue, and they establish functional interactions with non-transformed cells of the tumor microenvironment (TME) or niche. With tumor progression, cancer cells can affect the TME to adopt them for their needs. Autophagy in the TME is an important element of non-cancer cells’ fitness and their ability to support tumor growth [90].

Interestingly, cancer cells seem to stimulate autophagy in TME cells [90,91]. In turn, autophagy-dependent metabolite secretion by the TME is pivotal to maintain the metabolism and growth of tumor cells. In this regard, a high level of basal autophagy was reported in cancer-associated fibroblasts (CAFs) from patients with head and neck squamous carcinoma (HNSC) in comparison to normal fibroblasts from the same anatomical area, evidenced by the presence of high number of autophagosomes and increased level of LC3 [92]. Conditioned media from CAFs with mitigated autophagy (*BECN1* knockdown or CQ treatment), in contrast to conditional media from non-manipulated CAF, were not able to maintain growth of HNSC cells in vitro, indicating the importance of CAF autophagy-dependent secretion for HNSC progression [92]. Autophagy-dependent alanine secretion from stromal cells was demonstrated to be essential for the metabolism and growth of pancreatic ductal adenocarcinoma [93].

In hematologic malignances, special attention should be given to the metabolic features of leukemic cells, which allow them to adapt to the niches of the bone marrow (BM) [94]. There is evidence that microenvironmental autophagy may play an important role in AML chemoresistance [74,95]. As was mentioned above, *ATG7* knockdown in AML cells increases their chemosensitivity, and this effect is enhanced by concomitant knockdown of ATG7 in both AML and stromal cells [74]. The levels of autophagy and ATG5 expression are increased in mesenchymal stem cells (MSCs) derived from the BM of AML patients (AML-MSCs) as compared to healthy donors [95]. When autophagy is inhibited by 3-Methyladenine (3-MA) or *ATG5* silencing in AML-MSCs, their cell cycle is arrested in G1, and the expression of CXCL12, which is responsible for the interaction of leukemic and stromal cells in the BM, is significantly reduced. When leukemic cells are cocultured with these “autophagy-modulated” AML-MSCs, they are more sensitive to the genotoxic agents daunorubicin and doxorubicin in comparison to leukemic cells cocultured with unmanipulated AML-MSCs.

Overexpression of stromal ATG16L [96] and ATG10 correlates with lymphovascular invasion and lymph node metastasis in human oral squamous cell carcinoma and colorectal cancer, respectively [97].

In AML, leukemic stem cells (LSCs), which are largely responsible for chemotherapy resistance and disease relapse [98], exhibit an oxidative phenotype (are OXPHOS-dependent), while normal hematopoietic stem cells (HSCs) display a glycolytic phenotype [99,100]. It has been shown that the metabolism of AML blasts is mainly based on OXPHOS rather than on oxidative glycolysis [89], but blasts, unlike LSCs, can up-regulate glycolysis to compensate for the loss of OXPHOS [99,100]. Additionally, AML blasts can generate hypoxic conditions in the BM, which drives the transfer of functional mitochondria from stromal cells to leukemic blasts via tunneling nanotubes [101]. A healthy microenvironment is then paramount for the survival of both AML blasts and LSCs.

## 4. Metabolic Changes Caused by Modulation of Autophagy in Cancer Cells

Several studies that have manipulated ATGs and autophagy followed by an assessment of cell metabolism have been reported and will be discussed in this section. The reported data are summarized in Table 1 and illustrated in Figure 1 (lower panel).

### 4.1. Metabolic Changes Caused by Autophagy Induction

When autophagy is induced by treatment of AML cell lines with the mTORC1 inhibitor rapamycin, glucose uptake is reduced as expected [102]. Similarly, glycolytic activity is reduced in the Panc-1 pancreatic cancer cell line treated with the rapamycin analogue everolimus [103].

### 4.2. Metabolic Changes Caused by Knockdown or Inhibition of ATGs

*ATG* silencing or pharmacologic inhibition of the respective proteins are common experimental approaches to study the role of ATG proteins in cancer cell metabolism. The following are compounds frequently used in experimental models to modulate autophagy at various stages.

Specific and potent autophagy inhibitor-1 (spautin-1) suppresses the deubiquitination activity of ubiquitin-specific peptidase 10 (USP10) and USP13 [104]. Inhibition of deubiquitinases by spautin-1 leads to ubiquitination and degradation of VPS34 and BECN1, both of which are critical regulators of phagophore formation in early autophagy [105,106].

3-MA blocks autophagosome formation via the inhibition of the class III PI3K complex [107].

Here, we provide details regarding the metabolic changes induced by the pharmacological blockade of several ATGs at different stages of autophagy in experimental models in vitro (Table 1, Figure 1).

The prevention of phagophore nucleation through the deletion of *BECN1* or pharmacologic inhibition of the class III PI3K complex by 3-MA attenuates lipid degradation, reduces FAO and decreases the basal and ATP-linked oxygen consumption rate (OCR, indicator of mitochondrial respiration) in the acute myeloblastic leukemia (AML) cell line MOLM14 [85]. In cell lines derived from solid tumors (colon and gastric cancers), *BECN1* ablation drastically decreases OXPHOS and lipid degradation and shifts cellular metabolism towards aerobic glycolysis, with increases in glucose uptake and lactate production [86,87]. Strengthening the idea of the shift from OXPHOS to aerobic glycolysis, BECN1 inhibition by Spautin-1 reduces OCR and suppresses mitochondrial complex I activity in a human fibrosarcoma cell line [108]. 3-MA causes down-regulation of genes involved in FAO and fatty acid transportation in a human hepatocellular carcinoma cell line [109].

Remarkably, deletion of *ATG3*, which is necessary for LC3 lipidation, or *ATG12*, which, in complex with ATG5, is required for the elongation of phagophores, also causes FAO reduction but increases glucose uptake and consumption in AML cell lines [85,110].

When a hepatocellular carcinoma cell line was transfected with inactive ATG4 (ATG4B^C74A^), compromising the first step of LC2 conjugation, it conserved a diminished FAO and lipid catabolism [109]. Similarly, deletion of *ATG5* and *ATG7* favors the maintenance of an increased level of glycolysis and glycolytic capacity in human cancer cell lines. In particular, the glycolytic enzyme HK2 level increases after the deletion of *ATG5* in liver cancer [111], while *ATG7* knockdown increases PKM2 phosphorylation [80]. Notably, tyrosine phosphorylation decreases PKM2 enzymatic activity [112].

In pancreatic adenocarcinoma cells, *ATG7* silencing increases glutamine consumption and decreases TCA cycle intermediate levels under normal conditions. However, with glutamine deprivation, this deletion further decreases intracellular glutamine levels, demonstrating that autophagy is necessary for the maintenance of intracellular glutamine levels and provides glutamine to support anaplerosis of the TCA cycle [113]. Furthermore, whereas *ATG7* deletion leads to impaired autophagy and favors glycolysis with lactate production (Warburg effect) [80,111], ATG7 overexpression inhibits the Warburg effect in the HeLa cell line by suppressing PKM2 phosphorylation [80].

Although several aforementioned studies evidence that ATG impairment causes metabolic shift toward glycolysis, contrasting data have also been reported. For example, spautin-1 reduces glucose uptake and the activity of HK2 in the human lung cancer A549 cell line [114]. There are also reports that *ATG7* deletion causes decreased LDH activity and glycolytic capacity in the MDA-MB-231 cell line [115] and decreases glucose uptake and lactate production in the human CML cell line K562 [116]. Deletion of *FIP200* (from the ULK complex) in mammary tumor cells derived from female C57BL/6 mice decreases glucose uptake and intracellular lactate production [117].

Similar findings have been described in the TAM-resistant MCF7 cell line, where AMPKα1 knockout impairs estrogen receptor-induced FAO [118].

### 4.3. The Blockade of the Autophagic Flux Alters the Metabolism of Human Cancer Cells

The autophagic process implies the need for the autophagosome to remain completely formed, enclosing the content to be degraded by the lysosome. Autophagosome–lysosome fusion can be prevented pharmacologically by using the lysosomotropic agent CQ [119] or Bafilomycin A1 (Baf A1), which disrupts autophagic flux by inhibiting both V-ATPase-dependent lysosome acidification and Ca-P60A/SERCA-dependent autophagosome–lysosome fusion [44,120]. When autophagic flux is blocked (autophagosome–lysosome fusion arrest), a large number of autophagosomes can accumulate, which causes the cancer cell to embark on the path of cell death, especially if autophagy was enhanced to survive in adverse environmental conditions.

An interesting link between HK2 levels and autophagic flux has been reported in liver cancer cell lines: cells with low autophagic flux exhibited high expression of HK2 and a high glycolysis phenotype, while cells with high autophagic flux acquired a low-glycolysis phenotype because HK2 is degraded by p62/SQSTM1-mediated autophagy [111] and CMA [121].

In human cholangiocarcinoma and colon and pancreatic cancer, CQ attenuates lipid degradation [86] with decreased levels of TCA cycle intermediates and increased glutamine consumption [113]. Additionally, CQ decreases the activity of glucose-6-phosphate dehydrogenase (G6PDH) [122], an enzyme that catalyzes the first step in the pentose phosphate pathway (a metabolic pathway parallel to glycolysis); in turn, a decrease in G6PDH activity favors the use of glucose-6-phosphate (G6P) in glycolysis.

Similarly, Baf A1 alters glucose and lipid metabolism, increases glucose uptake and lactate production in hepatocellular and ovarian carcinoma [111,123] and decreases basal and maximal OCR in patient-derived AML blasts or AML cell lines [124].

**Table 1 cells-12-02486-t001:** Modulation of autophagy and metabolic changes in cancer cells.

Autophagy Step	Strategy	Model *	Effect on Metabolism	Reference
Autophagy induction	Rapamycin	U937 and NB4 cell lines (acute myeloid leukemia)	Decreased glucose uptake	[102]
	(−) *AMPKα1*	TAM-resistant MCF7 cell line (breast adenocarcinoma)	Decreased expression of proteins that promote FAO through estrogen receptor	[118]
Initiation and phagophore nucleation	(−) *BECN1*	DLD1 cell line (colon cancer)	Attenuated lipid degradation	[86]
	(−) *BECN1*	SGC C-7901 and MGC-803 cell lines (gastric cancer)	Increased glucose uptake and lactate secretion Reduced citrate and fumarase level Shift from OXPHOS to glycolysis	[87]
	(−) *BECN1*	MOLM14 cell line (acute myeloid leukemia)	Attenuated lipid degradation Decreased basal OCR and ATP-linked OCR	[85]
	3-MA	Huh7 cell line (hepatocellular carcinoma)	Decreased intracellular ATP and β-hydroxybutyrate levels Down-regulation of genes involved in FAO and fatty acid transportation	[109]
	3-MA	MOLM14 and U937 cell lines (acute myeloid leukemia)	Attenuated lipid degradation Increased triglyceride levels Reduced FAO Decreased basal OCR and ATP-linked OCR	[85]
	Spautin-1	A549 cell line (lung cancer)	Decreased HK2 levels and glucose uptake	[114]
	Spautin-1	HT1080 cell line (fibrosarcoma)	Reduced OCR and suppressed mitochondrial complex I activity	[108]
Phagophore elongation	(−) *ATG3*	THP-1 and MV4-11 cell lines (acute myeloid leukemia)	Increased levels of fumarate and succinate Increased basal OCR and ATP-linked OCR Increased glucose uptake, glucose consumption and lactate production and decreased lactate excretion	[110]
	Inactive ATG4 (ATG4B^C74A^)	Huh7 cell line (hepatocellular carcinoma)	Decreased intracellular ATP and β-hydroxybutyrate levels Down-regulation of genes involved in FAO and fatty acid transportation	[109]
	(−) *ATG5*	SMMC7721 cell line (hepatocellular carcinoma)	Enhanced glucose consumption and lactate production Increased HK2 levels	[111]
	(−) *ATG7*	MDA-MB-231 cell line (breast cancer)	Decreased LDH activity, decrease in glycolytic capacity	[115]
	(−) *ATG7*	8988 T cell line (pancreatic cancer)	Increased glutamine consumption Decreased intracellular glutamine levels under glutamine deprivation conditions Decreased TCA cycle intermediates	[113]
	(−) *ATG7*	SMMC7721 cell line (hepatocellular carcinoma)	Increased glucose uptake and lactate production	[111]
	(−) *ATG7*	HeLa cell line (cervical carcinoma)	Increased PKM2 phosphorylation Increased glucose consumption and lactate production	[80]
	(+) *ATG7*	HeLa cell line (cervical carcinoma)	Reduced PKM2 phosphorylation Inhibition of the Warburg effect	[80]
	(−) *ATG7*	K562 cell line (chronic myeloid leukemia)	Decreased glucose uptake and lactate production Increased extracellular glutamate from transamination of α-ketoglutarate	[116]
	(−) ATG12	MOLM14 cell line (acute myeloid leukemia)	Reduced FAO	[85]
Cargo sequestration	(−) *p62*/*SQSTM1*	SMMC7721 cell line (hepatocellular carcinoma)	Increased HK2 levels	[111]
Fusion with lysosome	CQ	SW480 and DLD1 cell lines (colon cancer)	Attenuated lipid degradation	[86]
	CQ	8988 T and MIAPaCa2 cell lines (pancreatic cancer)	Increased glutamine consumption Decreased intracellular glutamine levels under glutamine deprivation conditions Decreased TCA cycle intermediates	[113]
	CQ	QBC939 cell line (cholangiocarcinoma)	Decreased glucose-6-phosphate dehydrogenase activity	[122]
	CQ	Primary chronic myeloid leukemia CD34^+^ cells	Increased levels of the TCA cycle intermediates (α-ketoglutarate and glutamate)	[116]
	Baf A1	BEL-7402; Huh7/SMMC7721 cell lines (hepatocellular carcinoma)	Increased glucose uptake and lactate production Increased HK2 levels	[111]
	Baf A1	BEL-7402 and HO-8910 cell lines (hepatocellular carcinoma and ovarian carcinoma, respectively)	Pathways related to glucose or lipid metabolism were altered	[123]
	Baf A1	Patient-derived AML blasts and MOLM-13 cell line (acute myeloid leukemia)	Decreased basal and maximal OCR	[124]

(−) knockdown; (+) overexpression; *: all models are of human origin; 3-MA: 3-methyladenine; Baf A1: bafilomycin A1; CQ: chloroquine; FAO: fatty acid β-oxidation; HK2: hexokinase II; LDH: lactate dehydrogenase; OCR: oxygen consumption rate; OXPHOS: oxidative phosphorylation; PKM2: Pyruvate kinase M2; TCA: tricarboxylic acid.

Taken together, the data presented in Section 4.1, Section 4.2 and Section 4.3 indicate that the silencing of the different ATGs as well as the pharmacological blockade of different phases of autophagy drastically alter the metabolism of cancer cells. In many types of tumors, a switch to the glycolytic phenotype and decrease in the use of fatty acids was observed (Table 1, Figure 1). Since contrasting data are also reported, a more detailed systemic analysis of the complex relationship between metabolism and autophagy in different cancer types at different stages of disease progression is needed.

## 5. Therapeutic Implications of Autophagy and Metabolism

### 5.1. How Cross-Talk between Autophagy and Cell Metabolism Affects the Drug Sensitivity of Tumor Cells

Most anticancer strategies based on autophagy modulation (mostly inhibition) are limited to monitoring cell viability. In our opinion, monitoring the concomitant metabolic changes is important, so respective data are listed in Table 1.

A dual strategy to improve the drug sensitivity of cancer cells is to simultaneously target metabolic pathways and autophagy (Table 2). Below, we present some outcomes of this strategy. Inhibition of glycolysis with 2-Deoxy-D-glucose (2-DG), a glucose analog that inhibits the function of HK2 and G6P isomerase [125], in combination with the autophagy inducer rapamycin prevents the induction of cell death caused by 2-DG in pancreatic and breast cancer [126], as well as neuroblastoma and colon carcinoma [127] cell lines. Controversially, HK2 inhibition with 3-Bromopyruvate (3-BrPA) and autophagy induction with rapamycin decrease cell proliferation with apoptosis induction in neuroblastoma cell lines [128].

Glycolysis inhibition with 3-BrPA, 2-DG or the pyruvate dehydrogenase kinase inhibitor dichloroacetate (DCA) in combination with phagophore nucleation inhibitors (3-MA or spautin-1) decreases cell viability and increases cell death in breast cancer [129], colon adenocarcinoma [108,130], nasopharyngeal carcinoma [131], pancreatic cancer [126], fibrosarcoma [108], melanoma [126,129,132] and stem-like population of human glioblastoma [133] cell lines.

Similarly, the glycolysis inhibitors 3-BrPA, 2-DG, lonidamine and DCA demonstrate greater cytotoxicity in *ATG7* [126,129,130] or *ATG5* [111] knockdown cancer cell lines than in the corresponding wild-type cell lines, as was evidenced by decreased viability and proliferation and increased cell death.

Consistent with this additive or synergistic effect of the pharmacological inhibition of glycolysis and inhibition of the initial steps of autophagy, blocking the autophagic flux with CQ in combination with the glucose transporter (GLUT) inhibitor silibinin enhances apoptosis induction in glioblastoma cell lines [134] and, likewise, 3-BrPA or 2-DG in breast cancer [129] and glioblastoma [135] cell lines, respectively.

On the other hand, the use of metformin, a drug that inhibits HK2 and mitochondrial complex I [136,137], in co-treatment with autophagy inhibitors rescues cells from metformin-mediated cytotoxicity: AMPK inhibitor compound C or AMPK knockdown decreases growth inhibition and cell cycle arrest in B-lymphoma and T-lymphoma cell lines [138]. 3-MA reduces apoptosis and growth/viability inhibition of B-lymphoma and T-lymphoma [138] and gastric cancer [139] cell lines. Spautin-1 enhances the inhibition of colony formation of BRCA1-deficient mammary tumor cells [140]. In contrast, inhibition of autophagy with 3-MA, CQ or *BECN1* knockdown reduces metformin-mediated cytotoxicity (viability and apoptosis) in the Ishikawa cell line (endometrial adenocarcinoma) [141].

Autophagy induction with rapamycin decreases the apoptosis and necrosis caused by the mitochondrial complex I inhibitors rotenone [142] or paraquat [143] in neuroblastoma cell lines. In contrast to this, 1-methyl-4-phenylpyridinium (MPP^+^, mitochondrial complex I inhibitor) [143] or doxycycline (inhibitor of mitochondrial biogenesis) [144,145] in combination with rapamycin increases cell death and decreases the cell proliferation of neuroblastoma [143] and glioblastoma [145] cell lines.

Additionally, depending on the cellular model and strategy used, the inhibition of autophagy in combination with drugs that alter OXPHOS by inhibiting mitochondrial complex I demonstrates different effects. Inhibition of ATG5-dependent autophagy through overexpression of a dominant negative form of ATG5 (dnATG5) in a neuroblastoma cell line increases cell death induced by paraquat or MPP^+^ [143]. CQ reduces necrosis induced by mito-lonidamine (which also inhibits mitochondrial complex I) in a lung adenocarcinoma that metastasized to the brain cell line [146].

Inhibition of glutaminolysis by glutaminase (GA) inhibitors (compound 968 or bis-2-(5-phenylacetamido-1,3,4-thiadiazol-2-yl)ethyl sulfide (BPTES)), in combination with suppression of autophagy by CQ or Baf A1, results in a decreased viability and increased apoptosis in pancreatic [113], non-small cell lung [147,148], colorectal [149] and breast [148] cancer. In the latter model, co-inhibition of glutaminolysis and FAO reduces cell viability [148].

Finally, inhibition of fatty acid synthase (FAS) with orlistat in combination with 3-MA shows a cytotoxic effect in a pancreatic cancer cell line [150], while in epithelial ovarian cancer cell lines, 3-MA reduces the loss of cell viability induced by orlistat [151]. A similar effect is caused by another FAS inhibitor, emodin [152], where blocking autophagy with CQ reverses the decrease in cell migration and invasion caused by emodin [153]. In addition, the carnitine palmitoyltransferase (CPT1) inhibitor etomoxir (FAO inhibition) individually and in combination with CQ produces a disruption of spheroid structure and cell death in primary ovarian tumor tissue [154].

**Table 2 cells-12-02486-t002:** Cross-talk between autophagy and cancer cell metabolism.

Autophagy Step Altered	Strategy	Metabolic Pathway Altered	Strategy	Model *	Biological Effects	Reference
Autophagy induction	Rapamycin	Glycolysis	3-BrPA	SH-SY5Y and SK-N-SH cell lines (neuroblastoma)	Decreased cell proliferation Increased cell death (apoptosis)	[128]
	Rapamycin	Glycolysis	2-DG	1420 and SKBR3 cell lines (pancreatic and breast cancer, respectively)	Decreased 2-DG-induced cell death	[126]
	Rapamycin	Glycolysis	2-DG	SK-N-BE and RKO cell lines (neuroblastoma and colon carcinoma, respectively)	Prevention of 2-DG-induced apoptosis	[127]
	Rapamycin	OXPHOS	Rotenone	SH-SY5Y cells (neuroblastoma)	Decreased rotenone-induced apoptosis	[142]
	Rapamycin	OXPHOS	Paraquat	SK-N-SH cell line (neuroblastoma)	Decreased paraquat-induced necrosis	[143]
	Rapamycin	OXPHOS	MPP^+^	SK-N-SH cell line (neuroblastoma)	Increased MPP^+^-induced cell death	[143]
	Rapamycin	OXPHOS	Doxycycline	U251 and U373 cell lines (glioblastoma)	Decreased cell proliferation	[145]
	(−) AMPKα	Glycolysis/ OXPHOS	Metformin	Daudi and Jurkat cell lines (B-lymphoma and T-lymphoma, respectively)	Decreased metformin-mediated growth inhibition and cell cycle arrest	[138]
	AMPK inhibitor compound C	Glycolysis/ OXPHOS	Metformin	Daudi and Jurkat cell lines (B-lymphoma and T-lymphoma, respectively)	Decreased metformin-mediated growth inhibition and cell cycle arrest	[138]
Initiation and phagophore nucleation	3-MA	Glycolysis	3-BrPA	MDA-MB-231 and MDA-MB-435 cell lines (breast cancer)	Decreased cell viability	[129]
	3-MA	Glycolysis	DCA	LoVo cell line (colon adenocarcinoma)	Enhanced DCA-induced necrosis Decreased cell viability and proliferation	[130]
	3-MA	Glycolysis	2-DG	CNE-1 and CNE-2 cell lines (nasopharyngeal carcinoma)	Decreased cell viability and colony formation and promotion of apoptosis	[131]
	3-MA	Glycolysis	2-DG	1420 and MDA-MB-435 cell lines (pancreatic cancer and melanoma, respectively)	Increased 2-DG-induced cell death and sensitization to 2-DG	[126]
	3-MA	Glycolysis/ OXPHOS	Metformin	Ishikawa cell line (endometrial adenocarcinoma)	Reduced metformin-mediated cytotoxicity (viability and apoptosis)	[141]
	3-MA	Glycolysis/ OXPHOS	Metformin	Daudi and Jurkat cell lines (B-lymphoma and T-lymphoma, respectively)	Decreased metformin-mediated growth inhibition	[138]
	3-MA	Glycolysis/ OXPHOS	Metformin	AGS cell line (gastric cancer)	Decreased metformin-induced loss of cell viability	[139]
	3-MA	Lipolysis	Orlistat	PANC-1 cell line (pancreatic cancer)	Reduced cell viability Increased apoptosis (caspase-3)	[150]
	3-MA	Lipolysis	Orlistat	SKOV3 and A2780 cell lines (epithelial ovarian cancer)	Decreased orlistat-induced loss of cell viability	[151]
	Spautin-1	Glycolysis	2-DG	HT-29 and HT1080 cell lines (adenocarcinoma and fibrosarcoma, respectively)	Decreased cell viability	[108]
	Spautin-1	Glycolysis/ OXPHOS	Metformin	BRCA1-deficient mammary tumor cells	Enhanced metformin-mediated inhibition of colony formation	[140]
	Spautin-1	Glycolysis	2-DG	GBM8 cell line (glioblastoma)	Enhanced 2-DG-induced loss of cell viability Increased apoptosis	[133]
	(−) BECN1	Glycolysis/ OXPHOS	Metformin	Ishikawa cell line (endometrial adenocarcinoma)	Reduced metformin-mediated apoptosis	[141]
Phagophore elongation	(−) *ATG7*	Glycolysis	3-BrPA	MDA-MB-231 and MDA-MB-435 cell lines (breast cancer)	Decreased cell viability	[129]
	(−) ATG7	Glycolysis	DCA	LoVo cell line (colon adenocarcinoma)	Enhanced DCA-induced necrosis Decreased cell viability and proliferation	[130]
	(−) *ATG7*	Glycolysis	2-DG	1420 cell line (pancreatic cancer)	Increased 2-DG-induced cell death	[126]
	(−) *ATG5*	Glycolysis	2-DG 3-BrPA Lonidamine	SMMC7721 cell line (hepatocellular carcinoma)	Decreased cell proliferation	[111]
	dnATG5	OXPHOS	Paraquat MPP^+^	SK-N-SH cell line (neuroblastoma)	Increased paraquat/MPP^+^-induced cell death	[143]
Fusion with lysosome	CQ	Glycolysis	3-BrPA	MDA-MB-231 and MDA-MB-435 cell lines (breast cancer)	Decreased cell viability Induction of apoptosis/necrosis Enhanced pro-apoptotic Bax and Bak expression	[129]
	CQ	Glycolysis	Silibinin (Silybin)	A172 and SR cell lines (glioblastoma)	Enhanced silibinin-induced apoptosis	[134]
	CQ	Glycolysis	2-DG	U251 cell line (glioblastoma)	Induction of cytotoxicity	[135]
	CQ	Glycolysis/ OXPHOS	Metformin	Ishikawa cell line (endometrial adenocarcinoma)	Reduced metformin-mediated cytotoxicity (viability)	[141]
	CQ	Glutaminolysis	BPTES	8988 T and MIAPaCa2 cell lines (pancreatic cancer)	Decreased cell viability Induction of apoptosis	[113]
	CQ	OXPHOS	Mito-Lonidamine	H2030BrM3 cells (from brain metastases of H2030 cell line, lung adenocarcinoma)	Reduced Mito-Lonidamine-induced necrosis	[146]
	CQ	Lipolysis	Etomoxir	Primary ovarian tumor tissue	Disruption of spheroid structure and cell death	[154]
	CQ	Lipolysis	Emodin	HepG2 cell line (hepatocellular carcinoma)	Reduced emodin-mediated inhibition of migration and invasion	[153]
	CQ	Glutaminolysis	GA inhibitor-968	H1299 cell line (non-small cell lung cancer)	Enhanced 968-mediated cell growth inhibition	[147]
	CQ	Glutaminolysis	GA inhibitor-968	SW480 and SW620 cell lines (colorectal cancer)	Decreased cell viability Induction of apoptosis	[149]
	CQ	Glutaminolysis	GA inhibitor-968	MDA-MB-231 and HCC38 cell lines (breast cancer); NCI-H1838 cell line (non-small cell lung cancer)	Decreased cell viability	[148]
	Baf A1	Glutaminolysis	GA inhibitor-968	MDA-MB-231 cell line (breast cancer)	Decreased cell viability	[148]

(−) knockdown; *: all models are of human origin; 3-MA: 3-methyladenine; Baf A1: bafilomycin A1; CQ: chloroquine; 2-DG: 2-Deoxy-D-glucose; 3-BrPA: 3-Bromopyruvate; DCA: Dichloroacetate; BPTES: bis-2-(5-phenylacetamido-1,3,4-thiadiazol-2-yl)ethyl sulfide; MPP^+^: 1-methyl-4-phenylpyridinium; dnATG5: dominant negative form of Atg5; OXPHOS: oxidative phosphorylation; GA: Glutaminase.

Although the ultimate consequence of alteration (mainly inhibition) of various metabolic pathways in combination with inhibition of autophagy at different stages largely depends on the cell model evaluated and the drugs used, most of the data presented here point to the fact that simultaneous inhibition of autophagy and cellular metabolism is a promising strategy against cancer cells.

### 5.2. Combination of Drugs Targeting Autophagy and Metabolic Pathways as a Strategy That Can Improve Chemotherapeutic Protocols

Therapeutic compounds, targeting either metabolic enzymes or autophagy, show efficacy in multiple cancers at various stages of clinical trials, but chemoresistance still represents the major challenge in cancer treatment [6]. In this regard, currently available data indicate that the plasticity and chemoresistance of tumor cells is based on the cross-talk of autophagy and metabolic processes (glycolysis, FAO, glutaminolysis and OXPHOS). Thus, a combined therapy that simultaneously targets autophagy and metabolic pathways may represent a viable strategy to overcome chemoresistance.

Inhibition of glycolysis in various types of cancer can induce autophagy, which prevents cells from dying: glucose deprivation causes metabolic reprogramming towards mitochondrial OXPHOS in an autophagy-dependent manner, which allows the survival of multiple types of tumor cells, including pancreatic cancer (PANC-1), cervical cancer (HeLa) and lung adenocarcinoma (A549) [155]. In human glioblastoma, glycolysis inhibition leads to the induction of autophagy, senescence and escape from apoptosis [133].

Glucocorticoid resistance in acute lymphoblastic leukemia from T lineage (T-ALL) cell lines was shown to be related to the induction of a moderate level of protective autophagy (mitophagy), increased consumption of fatty acids and FAO after treatment with dexamethasone (Dex). Then, combined treatment of T-ALL with Dex and autophagy inhibitor CQ increased Dex cytotoxicity and made the Dex-resistant T-ALL Dex-sensitive [156].

Most tumors rely on aerobic glycolysis for ATP production (Warburg effect). As was mentioned above, some glycolytic enzymes are involved in the regulation of autophagy and maintenance of FAO and OXPHOS (Section 3). Not surprisingly, autophagy suppression results in up-regulation of glycolysis in many cancer types (Section 4). Thus, suppression of glycolysis may represent a valid strategy to improve the therapeutic response of cancer patients to autophagy inhibitors. The possible effectiveness of such a strategy was demonstrated in in vitro experiments, where inhibition of HK2 decreased the viability of autophagy-impaired liver cancer cells [111].

As mentioned above, the glycolytic enzymes PKM2 and PGK1 can activate *BECN1* via phosphorylation, while LDHA can do so via complex formation (Figure 2). It is noteworthy that deficiency of PKM2 compromises progenitor and AML cells, while LDHA deletion inhibits the function of HSCs, progenitor and AML blasts [157]. Association of BECN1 with LDHA induces pro-survival autophagy in a model of tamoxifen-resistant breast cancer [83], while overexpression of PKM2 has been linked to increased proliferation and metastasis [158] and its phosphorylation promotes tumor growth and the Warburg effect [111], regulated by ATG7 [80]. These effects could be explained by the fact that enzymes such as HK2 and LDHA are direct targets of oncogenic transcription factors [6]. In addition, LDHA has been associated with drug resistance in AML [131].

It has recently been demonstrated that FAO favored the increased biogenesis of mitochondrial membrane phospholipids, mediated by STAT3-acetylation, which in turn increased mitochondrial membrane potential and caused resistance to paclitaxel in breast cancer cells [159]. Generally, the transcription factor STAT3 (signal transducer and activator of transcription 3) can regulate autophagy in various ways, depending on its subcellular localization: (1) nuclear STAT3 regulates the transcription of various ATGs, including *BECN1*; (2) cytoplasmic STAT3 interacts with autophagy-related signaling molecules and inhibits autophagy; (3) mitochondrial STAT3 suppresses ROS-induced mitophagy [160]. This correlates with the overcoming of chemoresistance to paclitaxel in a model of lung adenocarcinoma through the inhibition of lipolysis by mercaptoacetate or etomoxir [161].

p62/SQSTM1 and LC3 expression is associated with tumor recurrence in oral squamous cell carcinoma [162], whereas p62/SQSTM1-LC3 interaction increases invasiveness in lung cancer and correlates with poor prognosis [163]. Similarly, LC3-II expression in colorectal cancer tissues is higher than in normal tissue [164]. In this sense, inhibition of p62/SQSTM1 causes autophagic cell death in adenocarcinomas and squamous cell carcinomas [165].

In a model of lung cancer, knockdown of *ATG5*, *BECN1*, *ATG7* and *p62/SQSTM1* decreases A549 cell invasiveness in the presence of cancer-associated fibroblasts (CAFs). In this model, even in the absence of CAFs, lung cancer cells exhibit an elevated autophagic flux that favors invasive ability, while blocking autophagic flux (CQ) completely inhibits this invasion [163].

Furthermore, ATG9 has been reported as a target in hypoxia-induced autophagy, so the inhibition of autophagy decreases cell proliferation and glioblastoma tumor growth in vivo [166], whereas in a model of B-cell leukemia, the knockdown of hVPS34 reduces cell proliferation and survival [167].

## 6. Conclusions

Under stressful conditions, cancer cells resort to autophagy as a strategy that allows the elimination of damaged organelles and recycling of metabolic blocks for survival and proliferation. This fact has provided the basis for the widely accepted view that inhibition of autophagy is an effective approach for anticancer therapy. This view, however, is not always correct. In this review, we provided experimental evidence that the suppression of ATGs in many types of tumors can lead to activation of the Warburg effect, characteristic for aggressive tumors. In particular, the overexpression of glycolytic enzymes such as HK2, LDHA and PKM2, which promote glycolysis and survival, was observed. Considering this adaptation mechanism, the use of different autophagy inhibition strategies should be carefully evaluated for each cancer type. To prevent chemoresistance, the possibility of a combination of autophagy inhibitors with the inhibitors of cellular metabolism, in particular inhibitors of glycolysis, should be considered.

## Figures and Tables

**Figure 1 cells-12-02486-f001:**
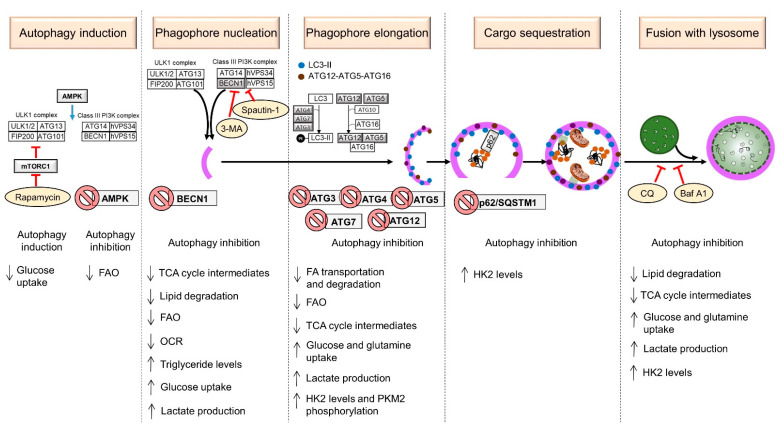
Metabolic alterations caused by modulation of autophagy in cancer cells. The stages of autophagy are indicated. The ATGs subjected to modulation are shown in gray. Silencing genes are labeled with a prohibition sign. Inhibitors are enclosed in yellowish ovals. Symbols are as follows: → (blue bold arrow), modulation; Ⱶ (red), inhibition; ↓, process down-regulated; ↑, process up-regulated.

**Figure 2 cells-12-02486-f002:**
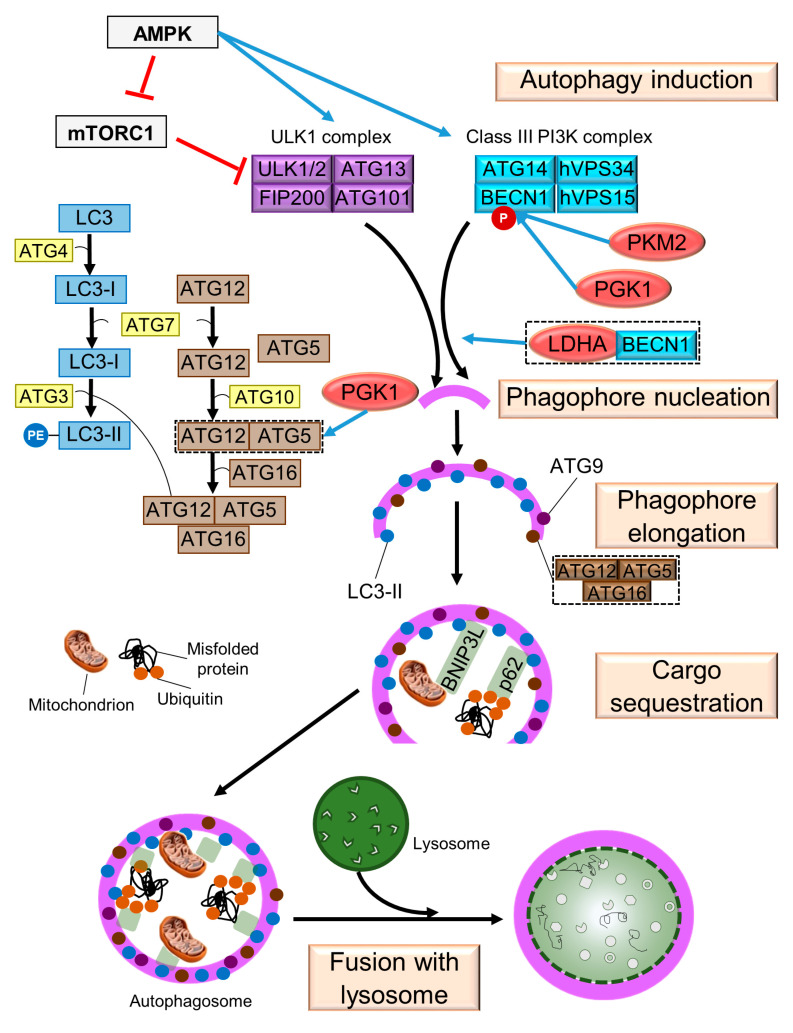
Glycolytic enzymes that favor autophagy in cancer cells. The stages of autophagy are indicated. Glycolytic enzymes are enclosed in red ovals. Symbols: → (blue bold arrow), modulation; Ⱶ (red), inhibition.

## Data Availability

Not applicable.

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
