# Peer review of "Dissecting the Role of Autophagy-Related Proteins in Cancer Metabolism and Plasticity"

_cells, 2023, doi:10.3390/cells12202486_

Round 1
Reviewer 1 Report
The review article titled “Dissecting the role of autophagy-related proteins in cancer metabolism and plasticity” is interesting and well written. In general, the article is well drafted and organized. However, there are a few minor concerns-
1) Introduction lines 31-34: The sentence is awkward and needs to be reconstructed.
2) Lines 45-46: The authors state that autophagy is regulated through different metabolic pathways, however, these pathways are not specified or discussed here.
3) Mechanism of autophagy in mammals is too detailed and exhaustive.
4) Discussion heading should be replaced with therapeutic implications of autophagy and metabolism.
5) The conclusion is too vague and lacks a future perspective on the topic.
Author Response
The review article titled “Dissecting the role of autophagy-related proteins in cancer metabolism and plasticity” is interesting and well written. In general, the article is well drafted and organized. However, there are a few minor concerns
Our response: We thank the reviewer for taking the time to review our manuscript and for kindly evaluating our work. Below are detailed responses and changes, highlighted in yellow, in response to critical suggestions.
1) Introduction lines 31-34: The sentence is awkward and needs to be reconstructed.
Done. The sentence has been rewritten for better understanding. Lines 31-35.
2) Lines 45-46: The authors state that autophagy is regulated through different metabolic pathways, however, these pathways are not specified or discussed here.
The content of this paragraph has been expanded. We also provide references for detailed reviews covering this topic. Lines 48-62.
3) Mechanism of autophagy in mammals is too detailed and exhaustive.
The general mechanism of autophagy has been rewritten in a more summerized form. Lines 104-159.
4) Discussion heading should be replaced with therapeutic implications of autophagy and metabolism.
Done
5) The conclusion is too vague and lacks a future perspective on the topic.
The conclusion has been expanded. Future perspectives are highlighted in the final sentences: “Considering this adaptation mechanism, the use of different autophagy inhibition strategies should be carefully evaluated for each cancer type. To prevent chemoresistance, the possibility of combination of autophagy inhibitors with the inhibitors of cellular metabolism, in particular, inhibitors of glycolysis, should be considered.” The graphical abstract also reflects this proposal for the development of therapeutic strategies in the future.
Reviewer 2 Report
The current review “Dissecting the role of autophagy-related proteins in cancer metabolism and plasticity” highlights the role of autophagy mechanism where they emphasize the proteins/genes involved in the formation of autophagosome. Further they talk about how upregulation/ downregulation of ATGs modulate the cancer. However, there are certain points need to be addressed.
Major comments –
1) The introduction needs to be concise. It would be good to add Drugs that are modulating autophagy which are currently used in clinical trials. (Line -43).
2) Please mention the metabolic pathways that are involved in autophagy. (Upregulates) Line -45.
3) From the discussion section please remove 5.1- “Cross-talk between autophagy and cell metabolism affect drug sensitivity of tumor cell”. 5.2- Combination of drugs targeting autophagy and metabolic pathways as strategy that can improve chemotherapeutic protocols.
4) Explain how autophagy aids in stress recovery (e.g., starvation).
5) Could the authors explain why targeting autophagy pathways in cancer is a more effective approach?
6) Discuss how autophagy is involved in inflammation?
Minor Comments -
1) Abbreviations must be written in full length when mentioned for the first time. (Example- PreB-ALL).
2) Line 473- Please check the spelling of Major.
3) There are numerous grammar errors, incorrect tense uses, and typographical errors.
Language editing is needed for consistency and flow of sentence.
Author Response
The current review “Dissecting the role of autophagy-related proteins in cancer metabolism and plasticity” highlights the role of autophagy mechanism where they emphasize the proteins/genes involved in the formation of autophagosome. Further they talk about how upregulation/ downregulation of ATGs modulate the cancer. However, there are certain points need to be addressed.
Our response: We thank the reviewer for his comprehensive review of our manuscript. Below are detailed responses and changes highlighted in yellow in response to suggestions.
Major comments
1) The introduction needs to be concise. It would be good to add Drugs that are modulating autophagy which are currently used in clinical trials. (Line -43).
Done. Lines 43-46
2) Please mention the metabolic pathways that are involved in autophagy. (Upregulates) Line -45.
Done. The content of this statement was expanded, pointing out reviews that have addressed this topic. Lines 48-62
3) From the discussion section please remove 5.1- “Cross-talk between autophagy and cell metabolism affect drug sensitivity of tumor cell”. 5.2- Combination of drugs targeting autophagy and metabolic pathways as strategy that can improve chemotherapeutic protocols
Considering the extension of the "discussion" section, we decided to keep both subtitles and replace the discussion heading with a more appropriate title, according to suggestion of another reviewer, but maintaining at all times the discussion of the information presented in the previous chapters. All this with the aim of not losing the content of this section
4) Explain how autophagy aids in stress recovery (e.g., starvation).
In this regard, a statement has been added in lines 51-62, in addition to what is contained in lines 73-75
5) Could the authors explain why targeting autophagy pathways in cancer is a more effective approach?
Comparing autophagy targeting and metabolism targeting, we do not believe that one is more effective than the other for all cancer types. In contrast, our suggestion reflected in the conclusion highlights the need to combine both strategies. Lines 521-532
6) Discuss how autophagy is involved in inflammation?
We understand the logic of this suggestion, since most human diseases, including cancer, have an inflammatory component. However, in considering the possibility of including this additional topic in this review, we concluded that the topic is too complex to be covered in a single paragraph or chapter. Whereas a detailed discussion will significantly increase the size of review and will lead away from the main topic, namely, what metabolic changes are caused by ATG modulation.
Minor Comments
1) Abbreviations must be written in full length when mentioned for the first time. (Example- PreB-ALL).
Done. Lines 193, 372, 413, 472
2) Line 473- Please check the spelling of Major.
Corrected
3) There are numerous grammar errors, incorrect tense uses, and typographical errors.
Done.
Comments on the Quality of English Language: Language editing is needed for consistency and flow of sentence.
English grammar and style have been corrected by an English-speaking colleague.
Reviewer 3 Report
The paper was well-organized and well-written. I only has a minor suggestion.
1. The gene names must be written in italics.
The language needs minor revision.
Author Response
The paper was well-organized and well-written. I only has a minor suggestion.
Our response: We thank the reviewer for taking the time to review our manuscript and for kindly evaluating our work. Below are detailed responses and changes, highlighted in yellow, in response to his/her comments.
1. The gene names must be written in italics.
Done. Lines 172, 178, 181, 195, 230, 235, 264, 269, 325, 330, 332, 333, 335, 340, 346, 348, 408. Tables 1-2
Comments on the Quality of English Language: The language needs minor revision.
English grammar and style have been corrected by an English-speaking colleague.
Round 2
Reviewer 2 Report
Authors have addressed all the comments and revised the paper accordingly.